# Genomic and Molecular Mechanisms of Goat Environmental Adaptation

**DOI:** 10.3390/biology14060654

**Published:** 2025-06-05

**Authors:** Ying Lu, Ruoshan Ma, Dongfang Li, Yuyang Gao, Zhengmei Sheng, Jinpeng Shi, Yilong Peng, Zhengdong Gao, Weidong Deng, Xiaoming He

**Affiliations:** 1 Yunnan Provincial Key Laboratory of Animal Nutrition and Feed, Faculty of Animal Science and Technology, Yunnan Agricultural University, Kunming 650201, China; yinglu_1998@163.com (Y.L.); maruoshan_2000@163.com (R.M.); dfli0927@163.com (D.L.); gaoyy5210@163.com (Y.G.); 18008847283@163.com (Z.S.); sjp020607@163.com (J.S.); zander_gao@163.com (Z.G.); 2 Faculty of Science, The University of Queensland, St Lucia, QLD 4072, Australia; pengyilong009@163.com; 3 College of Animal Husbandry and Veterinary Medicine, Yunnan Vocational College of Agriculture, Kunming 650201, China

**Keywords:** adaptive evolution, environmental adaptation of goats, epigenetic regulation, regulatory network

## Abstract

Goats are extremely adaptable animals that can thrive in a variety of environments, from hot deserts to cold plateaus and high altitudes. This article outlines how goats use genetic and epigenetic strategies to survive and grow in harsh conditions. We describe how specific genes help goats cope with high temperatures, low temperatures, and low-oxygen environments, as well as how molecular mechanisms such as DNA methylation and gene expression regulation fine-tune these responses. By summarizing recent scientific findings, we provide an integrative framework for understanding environmental adaptability in goats and propose specific goals, including the identification of key adaptive genes and their application in breeding programs to enhance the sustainability of precision breeding under climate change.

## 1. Introduction

Archaeological discoveries of goat DNA extracted from remains in the Zagros Mountains, dating back to the Neolithic period (around 10,000 BCE) [1], suggest that goats were domesticated from the wild goat (*Capra aegagrus*), making them one of the earliest domesticated animals in human history [2]. The evolution of goats has followed human migration patterns, marking the transition from hunting to pastoralism [3]. As a widely distributed livestock species, selective breeding by humans has driven the development of goat breeds specialized for meat, milk, fiber production, or those with multiple uses [4,5]. Currently, there are over 800 goat breeds worldwide, generally classified as meat (e.g., Boer), dairy (e.g., Saanen), and fiber (e.g., Angora) goats, with some multi-purpose breeds adapted to local environments.

One of the most striking aspects of goat evolution is their remarkable environmental adaptability, allowing them to thrive in diverse agricultural ecological conditions shaped by human civilization and migration patterns, whether in arid deserts [6] or cold plateaus [7]. This adaptability has led to a variety of physiological and genetic changes over time, enabling goats to survive in challenging environments, initially reflected in their morphological, physiological, and behavioral traits such as size, coat, and color [8]. Additionally, as ruminants, their unique digestive system with a multi-chambered stomach and metabolic mechanisms, coupled with adaptive foraging behaviors and microbial cooperation, enables them to efficiently digest and absorb fiber, making use of low-quality roughage [9]. Furthermore, goats adapt to diverse environments through highly flexible social behaviors, such as strong group cohesion to enhance stability and evade predators, helping them cope with seasonal and environmental changes [10]. Their reproductive cycles and mating strategies are also regulated by environmental factors, such as the suppression of reproduction in resource-scarce environments to ensure offspring survival [11]. These traits not only highlight goats’ exceptional ability to adapt to diverse environments but also reveal their evolutionary and ecological characteristics. They provide valuable resources and practical implications for agricultural production. Goats can effectively utilize marginal lands where other livestock may struggle to survive. They also consume a wider variety of plant species and exert relatively less pressure on plants and the environment. This contributes to improving overall agricultural productivity, resource efficiency, and biodiversity conservation [12]. Furthermore, goats emit only 20–25% of the methane produced by beef and dairy cattle, and improving goat management and grazing practices can promote environmental sustainability [13]. Therefore, goats are not only vital resources for human life but also play significant roles in cultural and economic activities.

Advances in genomics and epigenetics have made it increasingly important to explore the genetic mechanisms behind goat environmental adaptation. Recent studies have applied genome sequencing, transcriptomics, and functional validation to uncover how goats respond to extreme environments at the molecular level. However, many scientific questions remain unresolved, including the fine-scale genetic basis of specific adaptive traits, the synergistic interactions among adaptation-related genes, and the crosstalk between genomic and epigenetic regulation. Further research into the mechanisms of goats’ environmental adaptation is not only of significant practical importance for goat genetic breeding but also provides an ideal model system for understanding their evolutionary adaptation mechanisms.

## 2. Overview of Goat Genomics Research

### 2.1. Construction and Refinement of the Goat Reference Genome

In order to understand the genetic information of a species, the construction of a reference genome is fundamental to genomic research. As an important economic animal, the construction of the goat reference genome is crucial for revealing its genetic traits, disease resistance, and production performance (such as milk yield and meat quality). Moreover, it helps in discovering unique genetic features, protecting and utilizing the genetic resources of endangered breeds, and guiding molecular breeding and precision selection. Through high-throughput sequencing technologies and bioinformatics analysis, we are able to assemble high-quality goat reference genomes, providing a solid theoretical foundation and data support for subsequent gene function research, molecular breeding, and evolutionary analysis.

The initial goat genome construction was based on short-read sequencing technologies (such as the Illumina platform). In 2013, Dong et al. successfully performed de novo sequencing and assembly of the Yunnan black goat genome using high-throughput whole-genome sequencing, overcoming the bottleneck of relying on genetic maps to assemble large genomes to the chromosomal level [14]. Although second-generation sequencing is cost-effective and yields high-quality data, the short-read sequencing data struggle to address more complex genomes, especially the issues of repetitive sequences and structural variants [15]. As a result, long-read sequencing technologies such as PacBio and Oxford Nanopore have emerged, which can better span repetitive regions and improve genome continuity. Additionally, Hi-C and optical mapping technologies have been introduced as auxiliary techniques to provide support for the physical localization and chromosomal-level assembly of genomes. By combining these techniques, the assembly quality of the goat genome has been significantly improved.

In 2017, a high-quality goat genome assembly was achieved, primarily using PacBio sequencing data, in combination with optical mapping (BioNano (San Diego, CA, USA)) and Hi-C technologies for scaffolding. This resulted in 31 scaffolds, with a scaffold N50 of 87 M and a relatively low number of gaps [16]. To meet the requirement for assembly completeness, a gap-free genome, which involved completing all assembled regions on the chromosomes without gaps, was further combined with FISH, ChIP-seq centromere separation analysis, and other methods to accurately predict complex regions such as telomeres and centromeres. This achieved the complete assembly of at least one chromosome from telomere-to-telomere, known as the Telomere-to-Telomere (T2T) Genome [17]. In 2024, the first T2T genome assembly of the cashmere goat, including both autosomal and XY chromosomes, was completed. This assembly (T2T-goat1.0) achieved a core single-copy gene BUSCO assessment of 98%, indicating the high-quality assembly of the cashmere goat genome [18]. Table 1 presents representative goat genome assembly versions and evaluation metrics.

### 2.2. Features of the Goat Genome

Among domesticated ruminants, the goat genome is approximately 2.6 Gb [14], which is similar to closely related species such as sheep (*Ovis aries*, approximately 2.7 Gb) and cattle (*Bos taurus*, approximately 2.9 Gb) [23]. The goat genome contains a large number of short tandem repeats (STRs) and transposable elements (TEs). Differences in genome size are largely influenced by the length of non-coding regions and the content of repetitive sequences. These repetitive sequences account for 35–40% of the genome [16] and play a significant role in genome evolution, such as mediating gene recombination and regulating functional genes through transposons [24]. Goats have a chromosome number of 2*n* = 60, consisting of 29 pairs of autosomes and 1 pair of sex chromosomes. Studies on chromosome structure have shown that, although the karyotype of goats is highly stable, there are significant differences in the size of the X chromosome across different goat breeds. The differences between sex chromosomes and the structure of autosomes also carry important genetic information, influencing genetic variation and adaptive evolution [25,26].

The goat genome exhibits a high degree of conservation in overall structure compared to other ruminants [27]. Specifically, the synteny between goat and sheep genomes is high, with more than 85% of gene sequences directly aligning [28]. Approximately 22,000 functional genes have been predicted in the goat genome, most of which are located on the autosomes. In terms of functional genes, goats share various gene families with other ruminants in different aspects. In metabolism, the monoacylglycerol O-acyltransferase (MOGAT) gene family undergoes functional differentiation through gene duplication in ruminants, with *MOGAT2* and *MOGAT3* being particularly notable. These genes are uniquely highly expressed in the skin of ruminants and participate in lipid synthesis and metabolism [29,30], while in humans and other mammals, they are primarily expressed in the gut and liver [31]. Additionally, in the immune system, mucin-like proteins and proline-rich protein II families have been identified as two rumen-specific structural protein genes in ruminants, forming important components of their immune defense [30]. Other genes related to seasonal reproduction in goats, such as *KiSS-1* and *RFRP* genes [32,33], and genes like *PGR*, *FSHβ*, and *GnRHR* in ruminants, not only participate in regulating fertility but also ensure population continuity [34]. The shared nature of the metabolism, immune, and reproductive gene families between goats and other ruminants reflects their evolutionary strategy to adapt to herbivorous digestion and complex environments.

However, the goat genome shows unique rearrangements and variations in certain regions, which are related to important functional genes involved in disease resistance, fertility, and fiber production. Studies have found that the mucosal surface of the goat gastrointestinal tract contains mucus composed of mucins. Domestic goats have acquired the domesticated *MUC6* gene from the West Caucasian wild goat (*Tur*), which involves the integration of large sequence fragments to encode mucin and serve as a barrier for innate immunity, enhancing goats’ resistance to parasites [2]. Another study revealed that during domestication, gene duplication formed a multi-copy tandem-repeat LGALS9L gene family, with functional differentiation linked to immune adaptation during the domestication process [22]. *BMPR1B* and *BMPR2* regulate reproduction in goats and sheep via the hypothalamic–pituitary–gonadal axis [35]. The deletion mutation of BMPR1B increases the phenotype variation in goat litter size by 10.4% [28]. Wool traits are often related to specific rearrangements and variations in certain regions of the genome, affecting key factors such as hair follicle development, hair characteristics, and growth cycles, thereby influencing traits such as wool density, length, color, and resistance. The core gene family encoding wool fibers in both sheep and goats is the Trichocyte Keratin-Associated Proteins (KRTAPs) family. In sheep, 73 KRTAPs influence fiber structure, with variants like a 57 bp deletion in *KAP6-1* linked to thicker fibers [36] and disulfide bonding patterns affecting cashmere softness [37]. In goats, *KAP6-2* [38] and *KAP15-1* [39] are associated with fiber characteristics. In conclusion, structural variations and rearrangements in the goat genome shape its characteristics through gene duplication, deletion, or insertion. Furthermore, the goat’s remarkable adaptability is closely associated with specific genes and genetic variations within its genome.

## 3. Genetic Basis of Environmental Adaptation in Goats

The environmental adaptation of goats is achieved through complex genetic mechanisms that have evolved over thousands of years. These mechanisms include natural selection driven by environmental challenges and artificial selection from domestication millennia ago. The specific genetic variation patterns left by these two processes, known as functionally significant sequence variants, have shaped goat adaptability [40]. Goats exhibit rich genetic diversity, primarily reflected in their population’s geographical distribution and environmental adaptability, allowing them to survive in unique environments such as plateaus, deserts, and mountainous jungles. Goats living in different environments display distinct genotypes closely related to their adaptation to these environments.

### 3.1. Evolution and Survival of Goats in Different Environments

The goat genome contains various genes associated with environmental adaptation, which show different selection pressures under different geographical and climatic conditions, such as drought resistance, cold tolerance, and low-oxygen adaptation, leading to the diversity of goat breeds (Figure 1).

#### 3.1.1. Adaptation to Heat and Desert Environments

Goats exhibit exceptional adaptability to heat stress and desert environments, a trait that is becoming increasingly valuable as climate change models predict higher temperatures and drier conditions in many areas where goats are raised [41]. Goats living in hot climates typically have shorter coats and better heat conduction and dissipation abilities. Darker fur provides enhanced radiation shielding or improved thermal balance in terms of UV protection and heat absorption. Goats can also dissipate heat by increasing their breathing rate and increasing blood flow to the skin surface. Genes such as *KITLG* and *ASIP* are known to regulate pigmentation and have been associated with selective signals in desert-adapted breeds. These findings suggest that coat color is not only a phenotypic trait but also a genetically regulated adaptive feature. Heat-tolerant goat breeds such as the Alpine goat and West African Dwarf goat are capable of surviving in hot climates, showing good appetite and production performance [42]. In Saanen goats exposed to 37 °C for 6 h daily during late pregnancy, heat stress elevated cortisol levels and significantly upregulated *HSP70* and *ACTHR* expression in peripheral blood mononuclear cells, though milk production remained unaffected [43]. When Boerka goats were raised in tropical climates with varying humidity and topography, the study showed that goats raised in Bali had smaller body traits, including height, chest width, and chest depth, compared to those raised in Bangka Belitung Regency [44]. Research using Miseq sequencing on the gut microbiome of Chinese Hainan black goats revealed that the dominant gut microbiota consisted of *Firmicutes*, *Bacteroidota*, and *Pseudomonadota* [45]. Similarly, a study on Korean goats subjected to high temperatures found significant changes in metabolites such as butyrate, phenylacetate, and 2-oxoisocaproate in the rumen, and betaine and glucuronic acid in the blood. The study also found that *Prevotellaceae* and *Oscillospira* were enriched in the rumen. *Oscillospira* not only produces large amounts of butyrate and butyrate salts but also ferments complex plant carbohydrates, providing a carbon source for the animal [46,47]. In desert environments, Bedouin goats typically maintain energy balance by reducing their metabolic rate by 50%, despite only having 40% of the food supply [48]. To conclude, although high temperatures can stress the goat’s immune system and production performance, goats display robust survival and production potential through physiological and metabolic adaptation mechanisms such as adjusting respiration rates, metabolism, blood circulation, endocrine function, gut microbiota, and diet.

At the genomic level, heat stress adaptation in goats is closely associated with the activity of heat shock proteins and stress-related signaling pathways. For example, *HSP70* plays a central role in protein folding and cellular protection under elevated temperatures, while *ACTHR* is involved in stress hormone signaling. These genes have shown upregulated expression in goats under heat exposure, indicating their crucial roles in thermotolerance. Although current studies primarily focus on expression patterns, further research is needed to explore potential regulatory variants or selection signals in these genes that contribute to heat adaptation.

#### 3.1.2. High-Altitude Adaptation

The Tibetan Plateau, with an average elevation exceeding 4500 m, is not only one of the most challenging environments for animal survival but also an excellent model for studying high-altitude genetic adaptation. The extreme conditions, including low oxygen content, low pressure, cold temperatures, and intense UV radiation, present significant survival challenges for animals [49]. The Tibetan goat, through long-term natural selection, has developed stable physiological, biochemical, and morphological genetic traits [50]. One of the most prominent features of high-altitude environments is the low availability of oxygen. In such environments, oxygen transport and metabolism within animals are severely limited. In birds, it has been confirmed that species adapted to high-altitude hypoxia show significant changes in the affinity of hemoglobin for oxygen in their blood and muscles, while skeletal muscle metabolic pathways also undergo alterations [51].

Studies have shown that adult Tibetan goats maintain high levels of hemoglobin (Hb) for extended periods, enhancing their oxygen-carrying capacity. This is achieved by increasing myocardial hypertrophy and alveolar number or volume to meet the body’s oxygen demands and enhance gas exchange capabilities, while reducing heart rate within a normal range to minimize energy output and maintain sufficient blood flow to meet physiological needs [52]. These physiological features are likely influenced by the expression and regulatory variation in genes such as *EPAS1*, which enhances oxygen sensing and erythropoiesis, and *SIRT1*, which modulates cardiovascular responses under hypoxic conditions. Research also indicates that differences in the fiber diameter of Tibetan cashmere goats are likely related to their metabolic capacity, low-oxygen environment, and varying stress responses. Transcriptomic analysis of skin tissue revealed upregulation of *SLC2A1* for glucose metabolism and identified four proteins—AEBP1, VTN, GC, and GPR142—involved in matrix remodeling, tissue repair, vitamin D transport, and metabolic signaling, respectively, enriched in pathways related to hair fiber regulation [53]. Further integration of metabolomics, transcriptomics, and proteomics data to study variable cashmere phenotypes indicated that *DNMT3B* could be a key gene distinguishing cashmere fiber thickness, with differential proteins like HMCN1, LRP1, and CPB2 enriched in various KEGG pathways driving metabolic-related phenotypic differences in cashmere goats [54]. A comparative study of the physiological and biochemical indicators of local Changthangi goats and Black Bengal goats in high-altitude areas found that, initially, Black Bengal goats had significantly higher pulse rates and lower rectal temperatures and blood glucose levels compared to the local goats; however, over time, their indicators gradually normalized with environmental adaptation [55]. To summarize, the adaptation mechanisms of goats in high-altitude environments are reflected not only in regulating energy metabolism, optimizing the utilization of glucose and fatty acids, but also in the complex genetic and molecular regulation of multiple gene variations related to high-altitude adaptation.

#### 3.1.3. Cold Environment Adaptation

Goats’ ability to adapt to cold environments is influenced by several factors, including coat characteristics, temperature regulation, energy metabolism, and behavioral strategies. The density, length, and thickness of hair are key physiological traits affecting thermal insulation. Higher hair density and longer hair help reduce heat loss and enhance adaptability to cold environments [56]. Research on the wool fibers and diameter of Inner Mongolia cashmere goats identified genes related to hair follicle growth and development, such as *FGF12*, *SOX5*, and *EVPL*, which help cashmere goats better resist low temperatures [57]. Furthermore, goats generally regulate body temperature by adjusting basal metabolic rate (BMR), non-shivering thermogenesis (NST), and cold-induced thermogenesis (CIT). Special attention has been given to the unique characteristics of brown adipose tissue (BAT) in non-shivering thermogenesis. In cold environments, goats upregulate chi-let-7e-5p, which suppresses inhibitors of brown adipocyte formation, thereby supporting thermoregulation and energy metabolism [58]. Additionally, cold exposure has been found to induce BAT involvement in TCA cycle and fatty acid synthesis metabolic pathways, while *FGF11* activates EBF2 transcription factor activity, which subsequently activates *UCP1* expression and promotes BAT differentiation and thermogenesis [59,60]. These adaptive thermogenic responses are underpinned by the regulation of cold-responsive genes such as *UCP1*, *FGF11*, and *EBF2*, which drive brown adipocyte differentiation and energy metabolism during cold exposure. A cold stress experiment in newborn goats showed a significant increase in serum glucagon levels, while glucose and insulin concentrations were significantly reduced, activating skeletal muscle glucose metabolism pathways and the expression of genes involved in fatty acid and triglyceride synthesis [61]. Apart from temperature regulation and energy metabolism, goats in cold weather also exhibit behavioral adjustments, such as reduced respiratory rate [62] and increased indoor standing/walking activity [63]. The cold tolerance adaptation in goats involves multi-layered physiological, biochemical, and molecular regulatory mechanisms. Further research combining RNA-seq, proteomics, and gene editing technologies to explore the functions of key cold tolerance genes will aid in the improvement and precision breeding of cold-adapted goat breeds. To sum up, goats exhibit remarkable plasticity in adapting to different climatic conditions, reflecting the complex genetic mechanisms evolved over thousands of years for survival in diverse environments.

### 3.2. The Synergistic Mechanism of Immune Signaling Pathways and Environmental Adaptation

The immune system, as a crucial defense network for animals to cope with environmental challenges, plays a key role in goat responses to various ecological stresses. The immune response in animals is primarily mediated by both innate immunity and adaptive immunity. Innate immunity involves pathogen recognition receptors (such as Toll-like receptors (TLRs) and NOD-like receptors (NLRs)) that detect invading pathogens and activate rapid immune responses. In contrast, adaptive immunity primarily relies on the activation of B cells for antibody production and the differentiation and activation of T cells [64]. T cell receptors (TCRs) and helper T cells (Th cells) are involved in T cell-dependent immune responses. TCRs, which are specific receptors on T cell surfaces, primarily recognize specific antigens, while specialized Th cells are responsible for regulating and coordinating immune responses [65]. These two immune mechanisms work together through a complex signaling network, enhancing the goat’s ability to survive under different environmental pressures.

The goat immune system activates specific signaling pathways under environmental stress conditions to maintain homeostasis and defend against pathogen infections. Under heat stress conditions, the expression of mitochondrial fission protein 1 (FIS1) and dynamin-related protein 1 (DRP1) increases, while the expression levels of mitochondrial fusion proteins 1/2 are suppressed, leading to excessive mitochondrial fission in support cells and the inhibition of autophagic flux, which ultimately triggers cell apoptosis [66]. Recent studies have shown that melatonin can alleviate heat stress-induced support cell apoptosis by reducing ROS levels and restoring mitochondrial dynamic balance and autophagic flux [67]. Additionally, feeding high-starch degradable diets to dairy goats leads to hepatic metabolic dysfunction, increasing the likelihood of fatty liver disease. In tolerant individuals, gut microbiota induces a decrease in T helper 17 (TH17) cells and an increase in regulatory T cells by enhancing the synthesis of secondary bile acids, thus suppressing inflammation [68]. During pathogen infections, immune signaling pathways finely regulate autophagic activity to eliminate foreign pathogens. Research has shown that the goat immune system clears intracellular pathogens through autophagy mechanisms. TLR2-4 receptor genes activate the cAMP-PKA-NF-κB-ATGs pathway to enhance autophagy-associated signaling, while JNK/ERK signaling inhibits ATG5/ATG12 expression and negatively regulates autophagy [69]. This sophisticated immune regulation mechanism not only improves the goat’s resistance to pathogen infections but also reflects the immune system’s high plasticity and adaptability to different environmental conditions (Figure 2).

In sum, the immune system in goats works synergistically with environmental adaptation mechanisms through complex signaling networks. This synergy not only ensures the health and production performance of goats in different ecological environments but also highlights the core role of the immune system in animal environmental adaptation. Future studies on the interaction between immune signaling pathways and environmental adaptation mechanisms will help improve the precision of disease-resistant breeding and further enhance goats’ ability to adapt to environmental challenges.

### 3.3. Identification of Key Adaptive Genes in Goats

The genomic revolution has provided powerful tools for identifying genomic regions under selection, significantly enhancing our understanding of the genetic basis of goat adaptation. Whole-genome sequencing (WGS) and high-density single-nucleotide polymorphism (SNP) arrays provide comprehensive genetic information and efficient variation detection, deepening our understanding of the genetic mechanisms underlying environmental adaptation [70,71]. The detection of genetic marks (selective genomic features) left by natural and artificial selection events can be carried out using various statistical methods (Table 2), which are divided into population-based statistical methods, such as Cross-Population Composite Likelihood Ratio (XP-CLR), Cross-Population Extended Haplotype Homozygosity (XP-EHH), and Fixation Index (Fst), as well as genomic feature-based classification methods, such as Runs of Homozygosity (ROH), Latent Factor Mixed Models (LFMM), and Spatial Analysis Methods (SAMs) [72]. Among these, SAMs use a logistic regression model to study local adaptive traits related to the environment and spatial autocorrelation analysis [73]. LFMM is a new algorithm constructed based on population genetics, ecological modeling, and statistical analysis methods, and is used to screen local adaptive traits in the genome [74]. Of the two, LFMM focuses more on analyzing the association between genomic variations and environmental factors, while SAMs re primarily used for differential analysis of gene expression data.

Among these methods, population differentiation metrics such as Fst and composite likelihood approaches like XP-CLR are most commonly used due to their robustness in identifying genomic regions under divergent selection. Haplotype-based methods such as XP-EHH are also widely applied for detecting recent or ongoing selective sweeps, particularly in high-resolution genomic data sets.

In analyzing the genetic mechanisms of goat adaptive evolution, multidimensional statistical methods need to be integrated. For example, selective analysis using Fst and π has been employed to identify genes related to disease resistance and environmental adaptation in Yunnan goats, such as *DOK2*, *TIMM17A*, *MAVS*, *DOCK8*, and *LMOD1* [81]. Another study utilized LFMM, FST, XP-EHH, and SAM methods to identify 74 candidate genes related to climate-mediated selection in goats. Among these, *DENND1A*, *PLCB1*, and *ITPR2* were identified as relevant genes through all four methods [82]. Research on goat populations’ environmental adaptation under different climatic conditions is summarized in Table 3.

Furthermore, genomic variation plays a crucial role in the adaptation of goats to different environments, with variation types including single-nucleotide polymorphisms (SNPs), insertions and deletions (InDels), structural variations (SVs), and copy number variations (CNVs) [93]. For example, resequencing data have been used to identify genomic features and selective signals, revealing four high-frequency non-synonymous SNPs (*IFNGR1*:c.A733G; *SERPINB7*:c.A82G; *TLR2*:c.T47C; *TLR2*:c.A739G) associated with immunity, and a heat tolerance-related SNP (MTOR:c.G5507C) in Duan goats [94]. By performing exome sequencing comparisons between Tibetan cashmere goats (*Capra hircus*) and goats from lower-altitude regions, the study found synonymous SNPs in genes such as *SIRT1*, *ICAM1*, and *YES1*, and a novel non-synonymous mutation site Q579L in *EPAS1*, which was closely related to Hb levels. This suggests that *EPAS1* may be associated with goats’ adaptation to high-altitude environments [89,95]. Another study identified the *DSG3* gene’s significant role in Tibetan goats’ adaptation to high-altitude hypoxia, with three missense SNPs (R597E, T595I, G572S) and two synonymous SNPs in its CDS region. Polymorphism analysis showed a positive linear correlation between elevation and the frequency of mutated alleles, which was statistically significant [96]. Given the importance of genetic data in identifying adaptive genes and improving resistance in goats, gene editing technology provides a practical means. The study found that the innate inflammatory regulatory sequence was targeted and integrated into the promoter region of the lysozyme gene using the ISDra2-TnpB system, successfully constructing gene-edited dairy cows and goats with strong resistance to mastitis [68]. In addition to CRISPR/Cas9 gene knockout of the *MSTN* gene in sheep and goats, it was found that muscle development and growth rate were accelerated and the carcass was significantly larger compared to the wild type [97]. Gene editing technology can improve excellent goat traits by precisely adjusting related genes and can also enhance their resistance to environmental stress.

## 4. Synergistic Action of Genomic and Epigenetic Regulation

In addition to genetic selection, epigenetic processes also play an important role in responding to environmental factors by modifying gene expression [98]. Epigenetic regulation enables organisms to respond to external environmental changes such as temperature fluctuations, variations in nutrient levels, and exposure to harmful substances [99]. The core mechanisms of epigenetic regulation include DNA methylation, histone modifications, and post-transcriptional regulation mediated by non-coding RNAs (ncRNAs), which together contribute to the dynamic changes in chromatin structure and the regulation of gene transcriptional activity.

Among the various epigenetic mechanisms, DNA methylation is one of the most prominent epigenetic processes in goats. This process primarily occurs at the 5′ carbon of cytosine within CpG dinucleotides (CpG islands) and is catalyzed by DNA methyltransferases. Studies have found that the methylation levels of CpG islands in the growth hormone receptor (GHR) and growth differentiation factor-9 (GDF-9) promoters in Zaraibi goats were negatively correlated with their milk production performance (peak milk yield and total milk yield during the breeding season) [100]. By constructing a genome-wide DNA methylation map of the ovaries of Jining Grey goats, researchers identified methylation modification sites on genes such as *SERPINB2*, *NDRG4*, and *CFAP43*, which regulate litter size traits [101].

Histone modifications, another important epigenetic mechanism, occur when different groups are added to lysine or certain amino acid residues at the N-terminal of histones, altering the chromatin structure and subsequently influencing the binding of transcription factors and gene transcription efficiency [102]. A study investigating the regulatory mechanisms of steroidogenesis in goats found that H3K27me3 and H3K27ac were located in goat luteal cell nuclei. Adiponectin/AdipoRon regulates AMPK expression in the luteal cells and influences H3K27me3 levels through the Zeste 2 enhancer [103]. Additionally, gene expression can also be post-transcriptionally regulated by non-coding RNAs, including miRNA, siRNA, piRNA, circRNA, and lncRNA. Among them, lncRNA not only interacts with chromatin to influence gene expression but also participates in regulating cell differentiation and development [104]. In goats, a lncRNA expression profile identified 237 differentially expressed lncRNAs in the hypothalamus. These lncRNAs regulate target genes through cis- and trans-regulation, participating in sphingolipid signaling and GnRH signaling pathways to regulate goat sexual maturity and adapt to varying nutritional states and environmental stresses [105]. Through interactions among various ncRNAs and complex regulatory networks, goats exhibit potential adaptive mechanisms. For instance, in goats with different production capacities, differentially expressed lncRNAs and circRNAs in the pituitary regulate genes related to gonadotropin release and germ cell development (such as FSH and LH), significantly affecting fertility [106].

Moreover, epigenetic modifications not only exhibit plasticity during individual developmental stages but also play a role in the long-term adaptation of organisms to extreme environments. While most epigenetic changes are reversible and can be alleviated with environmental improvements to avoid unnecessary adaptive costs [107], studies have also shown that epigenetic information may be passed across generations, known as transgenerational epigenetics. This form of transmission is less stable than genetic mutations. For example, research on prenatal hypoxia exposure in sheep suggests that it enhances offspring’s tolerance to hypoxia, indicating that similar transgenerational effects may exist in goats [108] and highlighting the complex interaction between transgenerational genomic and epigenetic adaptation.

Genomic environment association analysis and selective scanning have been used to characterize the cashmere traits of Chinese cashmere goats, identifying new genes such as *ZEB1*, *ZNRF3*, and *MAPK8IP3* [109], which has further enriched understandings of the relationship between epigenetics and economic traits. The *TRAP1* promoter contains evolutionarily conserved hypoxia response elements, which respond to hypoxic conditions by activating HIF1α transcription and histone lactylation to maintain mitochondrial function and metabolic balance [110], demonstrating the complex interplay between genomic and epigenetic pathways in adapting to oxygen-limited environments. To sum up the above, goat gene expression regulation is driven by the synergistic action of multiple epigenetic mechanisms and genomic sequences. Epigenetics provides a flexible regulatory mechanism for organisms to adapt to external environmental changes while maintaining a constant genetic background. This additional layer of regulation in genetic control offers new theoretical support for understanding how animals maintain their competitive survival in complex ecological environments.

## 5. Research Strategies for Environmental Adaptation from a Multi-Omics Perspective

Although the previous article mainly discussed genomic and epigenetic mechanisms, there is increasing evidence that goats’ strong ability to adapt to the environment stems from multi-level mechanisms such as genetic diversity, physiological and metabolic regulation, and microbiome synergy. Therefore, from a multi-omics perspective, combining genomics, transcriptomics, and epigenomics, we can have a more comprehensive understanding of goat adaptability. Among them, environmental gradient–phenotype association studies and multi-omics dynamic monitoring and functional verification have become core strategies.

### 5.1. Environmental Gradient–Phenotype Association Study

Environmental gradient–phenotype association studies reveal the genetic basis of adaptive evolution by analyzing the relationship between environmental variables such as altitude and temperature in gradient changes and combining multi-omics data such as genome and transcriptome data. First, using high-density SNP chips or whole-genome resequencing data to conduct GWAS on populations with significant phenotype differences can identify candidate gene loci associated with environmental adaptability phenotypes [111]. Secondly, environmental variables can be combined with population genotype data to identify genes related to environmental adaptation using gene–environment association statistical models, i.e., environmental association genomics [82]. By comparing the genomes of goat populations at high altitudes (>4000 m) and low altitudes, the study found that the *PAPSS2* gene was introgressed from wild relatives (markhorn goats), significantly enhancing the Tibetan goat’s ability to adapt to hypoxia [112]. In addition, the properties of cashmere are also affected by environmental gradients. The goat T2T assembly found that the tandem duplication of the *ABCC4* gene on chromosome 12 in goats in high-altitude areas enhances the thermal insulation performance of cashmere by regulating the differentiation of hair follicle stem cells [18]. Finally, environmental gradients can also be combined with RNA-seq to reveal the association between gene expression levels and environmental phenotypes. The miRNA transcriptomes of six hypoxia-sensitive tissues of two goat populations at different altitudes were measured, and the results showed that miRNAs were involved in post-transcriptional regulation through the HIF-1, insulin and p53 signaling pathways, among which miR-106a-5p targeted *FLT-1* to negatively regulate angiogenesis [113]. Therefore, by covering sampling schemes at different altitudes, temperatures or humidity conditions, WGS combined with phenotypic measurements are performed on goat populations of the same breed, and GWAS or gene–environment association analysis is used to finely locate gene loci or regulatory regions that are strongly correlated with adaptive phenotypes, which can deepen the understanding of goat evolution and adaptation mechanisms and promote precision breeding and ecological protection.

### 5.2. Multi-Omics Dynamic Monitoring and Functional Verification

Under the same environmental gradient, genome, transcriptome, methylome, proteome and metabolome sequencing were performed on different tissue samples of goats to construct a dynamic data set of multiple time points and multiple tissues. With the help of CRISPR/Cas9 gene editing technology, functional knockout/knock-in verification was carried out in cells or model animals to reveal the causal relationship and regulatory mechanism of key genes.

Integrated liver transcriptome, metabolome, and intestinal metagenomic analysis revealed that Tibetan cashmere goats achieve adaptation to high-altitude environments by regulating liver gene expression (*PAK5*, *IP6K3*, *MFSD2A*) and glycolipid conversion metabolic pathways. Additionally, they improve their intestinal flora, including *Oscillibacter*, *Agrobacterium* and *Hyella* [114]. To evaluate the effects of heat stress on goat skin tissue, the study innovatively integrated the analysis of phenotypic traits, hair characteristics, cortisol levels, gene expression, skin histology and metagenomics. The results showed that Kodi Adu had a stronger ability to adapt to heat stress [115]. These examples illustrate the potential of multi-omics integration to uncover complex gene–environment interactions and support precision breeding efforts in challenging environments.

## 6. Conclusions

Goats exhibit exceptional adaptability to diverse environments, shaped by a combination of genetic, physiological, and epigenetic mechanisms. This review summarized current genomic insights into their environmental adaptation, highlighting key genes (*HSP70*, *EPAS1*, *UCP1*), immune responses, and epigenetic regulation. Advances in high-quality genome assemblies and multi-omics analyses have deepened our understanding of the molecular basis underlying these traits.

In the future, environmental gradient–phenotype association studies and multi-omics dynamic monitoring combined with functional verification will be important methods for revealing the adaptability of goat populations. Not only will candidate genes be screened efficiently, but the molecular evolution of adaptive mechanisms can also be elucidated from a spatiotemporal perspective. Algorithm innovation, data sharing, and construction of public databases for multi-omics data will continue to promote precision gene editing and molecular breeding, transforming research results into breeding tools for highly adaptable goat breeds.

## Figures and Tables

**Figure 1 biology-14-00654-f001:**
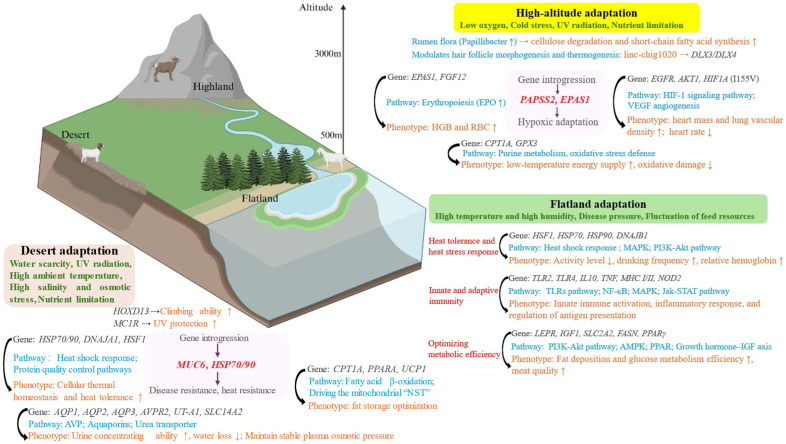
From genes to phenotypes: genetic mechanisms of adaptation to plateaus, plains and deserts. Arrows in the graph indicate an increase or decrease in genes or phenotypes.

**Figure 2 biology-14-00654-f002:**
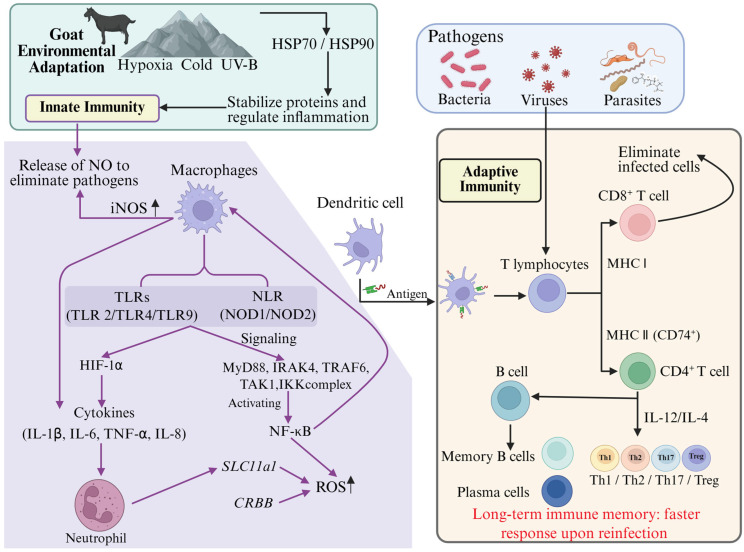
Adaptive immune mechanisms in goats. Abbreviations in the figure: TLRs—toll-like receptors, NLR—NOD-like receptors, ROS—reactive oxygen species, iNOS—inducible nitric oxide synthase, IL—interleukin, TNF-α—tumor necrosis factor-α, MyD88—myeloid differentiation primary response protein 88.

**Table 1 biology-14-00654-t001:** Research progress on genome assembly of goat.

Breed	Genome Versions	Sequencing Technology	Genome Size/Gb	Assembling Evaluation Metrics			Time	Reference
				Contig N50/Mb	BUSCO	Number of Gaps		
Yunnan black goat	CHIR_1.0	Illumina	2.8	0.15	95%	256,764	2013	[14]
*Capra aegagrus*	CapAeg_1.0	Illumina	2.8	0.15	90.7%	279,195	2015	[19]
Black Bengal goat	CVASU_BBG_1.0	Illumina	3.0	26.2	82.5%	3943	2019	[20]
Saanen dairy goat	Saanen_v1	Illumina, PacBio, Hi-C	2.7	46.2	98.3%	169	2020	[21]
San Clemente Island goat	ARS1	Illumina, PacBio, Hi-C	2.9	18.70	96%	649	2022	[16]
Bore Goat	ASM4458790v1	Illumina, PacBio HiFi, Hi-C	2.9	34.22	95.6%	165	2024	[22]
Nubian Goat	ASM4458793v1	PacBio HiFi	2.9	72.12	95.6%	119	2024	[22]
Inner Mongolia cashmere goat	T2T-goat1.0	PacBio HiFi, Ultra-long ONT, Bionano and Hi-C	2.9	100.8	98.0%	0	2024	[18]
Tibetan goat	ASM4646407v1	Illumina, PacBio, Hi-C	2.9	97.6	96.4%	243	2025	[7]

**Table 2 biology-14-00654-t002:** Modeling principles of commonly used selective signal detection methods.

Category	Statistical Methods	Detection Signal	Underlying Model	Individual/ Population Data	Reference
Population Differentiation	XP-CLR	Selective sweeps with allele frequency shifts	Composite Likelihood Model	population	[75]
	Fst	Allele frequency divergence between populations	Wright’s Island Model	population	[76]
Haplotype and LD-based Methods	XP-EHH	Completed selective sweeps	Extended Haplotype Homozygosity (EHH) Model	individual	[77]
	ROH	Artificial selection/demographic bottlenecks	Homozygosity Segment-based Analysis	individual	[78]
Environment Association Analysis	LFMM	Correlation between allele frequency and environment	Latent Factor Mixed Model (LFMM)	individual	[79]
	SAM	Genetic structure coupled with geographic space	Spatial Statistics Models (sPCA)	individual	[80]

**Table 3 biology-14-00654-t003:** Studies on the adaptation of major goat populations to climatic environments.

Adaptation Conditions	Species	Methods	Gene/Protein	Function	References
Tropical	Native Goats of Pakistan	SAM, Fst, θπ,	*KITLG*, *HSPB9*, *HSP70*, *HSPA12B*, *NBEA*	Thermotolerance	[83]
	Barbari goat	RT-PCR, Western Blot, Immunocytochemistry	iNOS, eNOS, cNOS, HSP70, HSP90	Improves heat stress and maintains cellular integrity and homeostasis in goats	[84]
	130 domesticated species (Abergelle, Garganica, etc.)	Fst, XPEHH, LFMM, SAM	*WDR75*, *SCN7A*, *PLCB1*	Thermotolerance	[82]
Desert	Iraq goat and Pakistan goat	Fst, θπ and Tajima’s D statistics	*KITLG*	Fur color change	[6]
	Capra nubiana	WGS, dN/dS ratio analysis, Fst	*ABCA12*, *ASCL4*, *UVSSA*	Participates in skin barrier protection	[85]
	Qaidam cashmere goats	WGS, Fst, Tajima’s D statistics	*CNGA4*, *Camk2b*	Enhance immune system function and resist adverse external factors	[86]
	Barki goats	Fst, iHS	*FGF2*, *GNAI3*, *PLCB1*	Heat resistance and melanin production	[87]
	Ugandan goat	Fst, hapFLK, ROH	*MTOR*, *MAPK3*, *HOXC12*, *IGF1*, *KPNA4*, *PPP1R36*	Involvement in the FAS pathway and regulation of HSP stress-induced heat tolerance	[88]
High altitude	Tibetan goat	Exome sequencing, SAM, Fst	*EPAS1*, *SIRT1*, *ICAM1*, *EDNRA*, *YES1*	Regulation of O_2_ utilization, inflammatory response, hemodynamics and cellular signaling	[89]
	Pashmina goat, Bakerwal goat	qRT-PCR, Whole-length amplification	*RBM3*	Activation of expression under low-temperature stress conditions	[90]
	Chinese indigenous goats	SNP Detection	*DSG3*	Mutation site polymorphism is strongly associated with hypoxia adaptation	[50]
	Dazu black goat	Thermographic evaluation, RNA-seq	*UCP1*, *CIDEA*, *PPARGC1a*	Regulation of BAT thermogenesis in goats	[91]
	Tibetan goat	iHS, ZHp, di	*CDK2*, *SOCS2*, *NOXA1*, *ENPEP*	Hypoxia adaptation	[92]

## Data Availability

No new data were created or analyzed in this study. Data sharing is not applicable to this article.

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
