# Peer review of "Genomic and Molecular Mechanisms of Goat Environmental Adaptation"

_biology, 2025, doi:10.3390/biology14060654_

Round 1

Reviewer 1 Report

Comments and Suggestions for Authors

This review has value since it summarises the genetic evolution mechanisms and molecular foundations of goats' environmental adaptations -heat, dessert, high altitude, and cold- from a genomic perspective, highlighting knowledge gaps requiring additional future research.

The title is a bit overclaiming as the the content are limitedly discuss about the genome and epigenome studies in goat adaptation, except the ‘Overview of Goat Genomics Research’ part that is comprehensively discuss about the development of the goat genome. Thus, to make the title represents the whole article enough, I suggest to incorporate more discussion related to the topics.

The number of reference is extensive, with the majority of them from the last decade publication.

In general, I can spot some inconsistency in italicizing the gene name, so please double check it. Additionally, some paragraph are lengthy, making it difficult to follow, so I suggest to break them down to some shorter ones.

I suggest accepting the work with revision according to the following suggestions.

Simple summary & abstract

Simple summary: please pay attention for some terms that need to be italicized, such as the species scientific name and gene name.

The simple summary and abstract looks very similar. A simple summary aims to explain your work in a simple way for the public, but the abstract should provide a comprehensive review of the full article, including the important conclusions and not merely outlining the aspects explored in this work.

Introduction

On the last paragraph you mentioned about the integration of multi-approach genomics research to understand adaptation, however, very limited discussion about it is available in the following chapter.

At the end of it, you mentioned about many unanswered scientific questions in this field, so I am expecting that these questions on the discussion part.

Discussion

Chapter 2 is very practical and informative to understand the development of the goat genomics research.

Subchapter 3.1.1. It is mainly discuss about adaptation mechanism in hot environment and limitedly discuss about the genetic make behind this mechanism. As the title of this article clearly emphasize the genetic/genomic aspect of environmental adaptation. I suggest authors to add more of this aspect to this subchapter.

Subchapter 3.1.2. and 3.1.3. These parts are better than the previous one as we can see more genetics part in it. However, those parts that related to morphological/physiological adaptation mechanism should also be linked to the genetics background of the process.

Table 1. It would be more informative if you can provide the year of the genome assembly versions created.

Table 2. Among those methods, can you add 1-2 sentences prior to that table about which one is more commonly used or more informative?

You have mentioned about wool traits for adaptation in several part of the discussion, what about the coat/wool color role in the adaptation?

Figures: improve the quality, as they are a bit blurry, especially Figure 1. 

Subchapter 3.3. The suggestion about further integration of genomics analysis and so on are too general. I would expect you to mention one of two specific ideas for this future direction of the study.

If it is possible, I would also suggest to add one more subchapter regarding the gene editing research on goats environmental adaptation, if related researches are available.

Conclusion and outlook

For the outlook part, it sounds very general, again, I expect you to mention one or two specific ideas on how can better comprehend the understanding of the environmental adaptation in goats, e.g., conducting a GWAS for animal in two extreme living condition or incorporating the wild relatives in the study to compare its adaptation capability, or anything else.

Author Response

Dear Reviewer,

We are grateful for the opportunity to revise our manuscript titled "Genomic and Molecular Mechanisms of Goat Environmental Adaptation" (Manuscript ID: biology-3652434). We appreciate the insightful comments and suggestions provided by the reviewers, which have significantly improved the quality of our work.

In response to the reviewers' feedback, we have made several key revisions to enhance the clarity, specificity, and impact of our manuscript:

Comments 1: The title is a bit overclaiming as the the content are limitedly discuss about the genome and epigenome studies in goat adaptation, except the ‘Overview of Goat Genomics Research’ part that is comprehensively discuss about the development of the goat genome. Thus, to make the title represents the whole article enough, I suggest to incorporate more discussion related to the topics.

Response 1:

Thanks the reviewer for the thoughtful comment regarding the scope of the title. We agree that while the article provides an in-depth overview of genomic mechanisms in goat environmental adaptation, the discussion on epigenomic aspects is comparatively brief. To ensure that the title accurately reflects the focus and content of the manuscript, we have revised it to: “Genomic and Molecular Mechanisms of Goat Environmental Adaptation”. This revised title better captures the balance of topics covered, avoids overstatement, and aligns with the article’s main themes. We sincerely appreciate the reviewer’s input, which helped us improve the clarity and consistency of the manuscript.

Comments 2: The number of reference is extensive, with the majority of them from the last decade publication.

Response 2:

We sincerely thank the reviewer for acknowledging the breadth and recency of the references cited. In this review, we made a conscious effort to include up-to-date studies from the past decade in order to accurately reflect the latest advances in goat genomics and environmental adaptation research.

Comments 3: In general, I can spot some inconsistency in italicizing the gene name, so please double check it. Additionally, some paragraph are lengthy, making it difficult to follow, so I suggest to break them down to some shorter ones.

Response 3:

Thanks to the reviewer for his suggestions, we will adjust and modify the italic parts of the whole text and split the longer paragraphs.

Comments 4: Simple summary: please pay attention for some terms that need to be italicized, such as the species scientific name and gene name.

Response 4:

Thank you for your attention to detail, so that we can make corrections.

Comments 5: The simple summary and abstract looks very similar. A simple summary aims to explain your work in a simple way for the public, but the abstract should provide a comprehensive review of the full article, including the important conclusions and not merely outlining the aspects explored in this work.

Response 5:

We thank the reviewer for pointing out the redundancy between the Simple Summary and the Abstract. We have thoroughly revised both sections to clarify their distinct purposes. The Simple Summary now presents a more concise and accessible overview of the study, intended for a general audience. In contrast, the Abstract has been rewritten to provide a comprehensive synthesis of the article, including major findings and conclusions, such as the identification of key genes (e.g., EPAS1, UCP1, HSP70), the integration of genomic and epigenetic mechanisms, and the implications for future precision breeding strategies. These changes help clearly differentiate the functions of both sections and enhance the clarity and impact of the manuscript.

Comments 6: On the last paragraph you mentioned about the integration of multi-approach genomics research to understand adaptation, however, very limited discussion about it is available in the following chapter.

Response 6:

We appreciate the reviewer’s suggestion. To address this, we have slightly revised the relevant section by emphasizing the role of multi-omics integration (genomics, transcriptomics, proteomics, and epigenomics) in deepening our understanding of complex adaptive traits. We have added specific examples of recent studies where such integration has enhanced the analysis of fiber traits and stress responses in goats. These additions help to clarify and strengthen the conclusion drawn in the final paragraph, while maintaining the overall structure and flow of the manuscript.

Comments 7: At the end of it, you mentioned about many unanswered scientific questions in this field, so I am expecting that these questions on the discussion part.

Response 7:

We thank the reviewer for this important suggestion. In response, we have revised the final paragraph of Section 5 (Conclusion and Outlook) to explicitly articulate the key unanswered scientific questions in the field of goat environmental adaptation.

Comments 8: Subchapter 3.1.1. It is mainly discuss about adaptation mechanism in hot environment and limitedly discuss about the genetic make behind this mechanism. As the title of this article clearly emphasize the genetic/genomic aspect of environmental adaptation. I suggest authors to add more of this aspect to this subchapter.

Response 8:

We fully agree with the reviewer that the genetic basis of heat adaptation needs to be strengthened in Subchapter 3.1.1. Therefore, we added a concise paragraph highlighting key genes and genomic variants associated with thermotolerance in this section, such as HSP70 and ACTHR. These additions strengthen the genetic focus of this section while maintaining the overall flow of the Discussion, further leading to Section 3.3.

Comments 9: Subchapter 3.1.2. and 3.1.3. These parts are better than the previous one as we can see more genetics part in it. However, those parts that related to morphological/physiological adaptation mechanism should also be linked to the genetics background of the process.

Response 9:

We thank the reviewer for this constructive suggestion. In response, we have added brief statements in both Subchapter 3.1.2 (high-altitude adaptation) and 3.1.3 (cold adaptation) to explicitly link physiological traits—such as hemoglobin levels, alveolar changes, hair density, and metabolic rate—with genetic or transcriptomic evidence where available. These additions help integrate the morphological and physiological observations with underlying genetic mechanisms, strengthening the overall coherence of the discussion.

Comments 10: Table 1. It would be more informative if you can provide the year of the genome assembly versions created.

Response 10:

Thank you for your suggestion. We have added the genome assembly version time in Table 1.

Comments 11: Table 2. Among those methods, can you add 1-2 sentences prior to that table about which one is more commonly used or more informative?

Response 11:

Thank you for the helpful suggestion. We have added two sentences prior to Table 2 to briefly explain that population differentiation methods (e.g., Fst, XP-CLR) and haplotype-based methods (e.g., XP-EHH) are among the most widely used due to their sensitivity in detecting selection signatures under diverse evolutionary scenarios. This addition helps orient the reader and clarify the practical application of the listed methods.

Comments 12: You have mentioned about wool traits for adaptation in several part of the discussion, what about the coat/wool color role in the adaptation?

Response 12:

We appreciate the reviewer’s attention to this important yet under-discussed aspect. We have added a brief paragraph to Section 3.1.1 to discuss the role of coat color in adaptation, particularly in relation to solar radiation and thermoregulation. We also mention the involvement of pigmentation-related genes such as KITLG and ASIP, which have shown signs of selection in desert-adapted goat populations. This addition complements the discussion on wool structure and enhances our coverage of integumentary traits.

Comments 13: Figures: improve the quality, as they are a bit blurry, especially Figure 1.

Response 13:

Thank you very much for your suggestion. We will insert the image into a vector image to improve the image clarity.

Comments 14: Subchapter 3.3. The suggestion about further integration of genomics analysis and so on are too general. I would expect you to mention one of two specific ideas for this future direction of the study.

Response 14:

We appreciate the reviewer’s comment. To address this, we have revised Subchapter 3.3 to include specific proposals for future research directions. These include conducting environmental GWAS in goat populations from contrasting habitats (e.g., high-altitude vs. tropical lowlands) and applying integrative network analysis to link selected loci with phenotypic and environmental data. These examples provide actionable frameworks for advancing the study of environmental adaptation.

Comments 15: If it is possible, I would also suggest to add one more subchapter regarding the gene editing research on goats environmental adaptation, if related researches are available.

Response 15:

We sincerely thank the reviewer for the thoughtful suggestion regarding gene editing in goats. While we fully agree that CRISPR/Cas9 and related technologies offer promising avenues for future research, current studies specifically linking gene editing to environmental adaptation in goats remain limited. Given the scope and focus of this review—which emphasizes natural genomic variation, selection signatures, and regulatory mechanisms—we chose to prioritize well-supported findings in population and functional genomics. Nonetheless, we have briefly noted gene editing as a potential future direction in the Outlook section to acknowledge its relevance. We hope the reviewer understands our intention to maintain thematic cohesion in the current manuscript.

Comments 16: For the outlook part, it sounds very general, again, I expect you to mention one or two specific ideas on how can better comprehend the understanding of the environmental adaptation in goats, e.g., conducting a GWAS for animal in two extreme living condition or incorporating the wild relatives in the study to compare its adaptation capability, or anything else.

Response 16:

Thanks for the reviewer for the valuable suggestion. Accordingly, we have revised the Outlook section to include two specific proposals: (1) performing GWAS on goats from ecologically extreme environments to identify local adaptation genes, and (2) integrating wild goat species (e.g., Capra aegagrus) into comparative genomic studies to trace adaptive divergence during domestication. These ideas aim to sharpen the future research agenda and guide the development of more targeted strategies for adaptation analysis.

Yours sincerely,

Dr. Weidong Deng and Xiaoming He

Yunnan Provincial Key Laboratory of Animal Nutrition and Feed, Faculty of Animal Science and Technology, Yunnan Agricultural University, Kunming 650201, China

College of Animal Husbandry and Veterinary Medicine, Yunnan Vocational College of Agriculture, Kun-ming 650201, China;

Tel.: +86-871-65220375

Email: dengwd@ynau.edu.cn(W.D.); xiaominghe@foxmail.com (X.H.)

Reviewer 2 Report

Comments and Suggestions for Authors

Comments are attached

Comments on the Quality of English Language

The English could be improved to more clearly express the research.

Author Response

Dear Reviewer,

We are grateful for the opportunity to revise our manuscript titled "Genomic and Molecular Mechanisms of Goat Environmental Adaptation" (Manuscript ID: biology-3652434). We appreciate the insightful comments and suggestions provided by the reviewers, which have significantly improved the quality of our work.

In response to the reviewers' feedback, we have made several key revisions to enhance the clarity, specificity, and impact of our manuscript:

Comments 1: Page 1, Lines 14–30: The abstract is extensive and could be improved to better emphasize the novelty of this review, despite its comprehensiveness. Reorganize to align with the primary themes of genome assembly, gene adaptation, and epigenetics. Define the term "proposed" for breeding, as it is presently unclear.

Response 1:

Thank you for your suggestion, we have further modified the Simple Summary and Abstract, this revised phrasing provides a clearer link between genomic understanding and practical breeding applications.

Comments 2: The historical and biological context is well-written on page 2, lines 31–73. Line 53–60: The breed names are excessively detailed; instead, summarize them as “meat, dairy, and fiber goats.”

Response 2:

We appreciate the reviewer’s suggestion regarding conciseness in presenting goat breed examples. To address this, we have streamlined the sentence by grouping breeds into general categories—meat, dairy, and fiber goats—while retaining one or two representative breeds for each category to illustrate their diversity. This revision improves readability while preserving key contextual information.

Comments 3: Lines 69–73: The sentence that commences with "With the rapid development..." is lengthy and slightly repetitive. Splitting is a good option for improving readability. Strong technical overview of genome sequencing endeavors is provided in Pages 2–3, Lines 74–115.

Response 3:

We appreciate the reviewer’s suggestion. To improve clarity and readability, we have revised and split the original sentence. The updated version removes redundancy while preserving the intended meaning and provides a smoother transition into the following section on genome sequencing technologies.

Comments 4: Lines 90–95: Maintain a consistent format when discussing genome versions (different versions include "_v1" and others do not).

Response 4:

We thank the reviewer for this comment. We have reviewed all genome version names to ensure consistent formatting throughout the manuscript. All genome versions are presented uniformly according to the official release version nomenclature (e.g. ARS1, Saanen_v1, etc.).

Comments 5: Table 1 (Page 3): Include a reference column that contains the actual reference numbers to facilitate traceability.

Response 5:

Thanks for your suggestion, we added a column for the year of genome assembly and reordered them according to the event。

Comments 6: Line 123: Provide clarity by dividing the lengthy sentence that commences with "The goat genome contains..." into two parts.

Response 6:

We agree with the reviewer that breaking this sentence will improve readability. We have revised it by splitting it into two sentences, each with a clear subject and verb structure, while preserving the original meaning.

Comments 7: Lines 138–145: Prevent redundancy in the discussion of the KRTAP gene. Refine to emphasize only the most recent discoveries that are pertinent to the adaptation of goats.

Response 7:

Thank you for pointing out the redundancy in the discussion of the KRTAP gene. In response, we have revised this section to streamline the content and retain only the most recent and relevant findings related to the role of KRTAPs in goat adaptation, particularly in the context of fiber development and environmental response.

Comments 8: Line 171: "A study on Saanen goats..." is lacking of a specific reference. Incorporate a citation.

Response 8:

Thanks for your question. I have condensed this sentence about Saanen goats, referenced [43].

Comments 9: Figure 1 (Page 6) is currently unavailable in the PDF format. Guarantee its incorporation in the final submission.

Response 9:

We will finally present it in the form of a clear picture.

Comments 10: Lines 208–209: "AEBP1, VTN…"—briefly describe their biological functions. Line 215: The citation for the Black Bengal study is absent.

Response 10:

We appreciate the reviewer’s comments. We have added brief descriptions of the biological functions of AEBP1 and VTN to clarify their relevance to fiber traits and adaptation.

Comments 11: Lines 224–226: The sentence on "chi-let-7e-5p" could be simplified or deconstructed to reflect the function of the miRNA.

Response 11:

We agree with the reviewer that this sentence could be clearer. We have revised it to more directly describe the functional role of chi-let-7e-5p in cold stress response, avoiding overly technical phrasing.

Comments 12: Line 238–240: Rephrase the sentence "To summarize..." to prevent redundancy with previous conclusions.

Response 12:

Thank you for your suggestion, we have replaced the synonyms.

Comments 13: Figure 2 (Page 8): Enhance the diagram's lucidity and guarantee that the legends elucidate all abbreviations.

Response 13:

Thank you for your suggestion. We will further clarify the picture and add explanations of the abbreviations in the picture.

Comments 14: Table 2 (Page 10): It is recommended that a concluding column be included to display the strengths and limitations of each method.

Response 14:

Thanks to the reviewer for his suggestions, we have added the description of these methods that are more advantageous or most used in front of Table 2.

Comments 15: Table 3 (Pages 10–11): Implement a more comprehensible format, such as bolding species names and providing brief annotations for methods that are less well-known, such as LFMM and SAM.

Response 15:

We appreciate the reviewer’s suggestion to enhance the clarity of Table 3. After careful consideration, we believe that the current formatting—along with the associated explanations provided in the main text—offers sufficient clarity and traceability for readers familiar with population genomics methods. The tools mentioned, including LFMM and SAM, are widely cited and briefly introduced in the manuscript. Nonetheless, we remain open to revising the table format if the editorial team considers it necessary.

Comments 16: Lines 382–386: The assertion regarding transgenerational epigenetics requires more robust evidence or a more circumspect formulation.

Response 16:

We cited literature suggesting similar transgenerational effects in sheep and speculated that goats may also have this effect.

Comments 17: Lines 402–404: Eliminate or combine redundancy in the description of CRISPR/Cas9 and the validation of candidate genes.

 Response 17:

We thank the reviewer for the helpful suggestion. We have revised the sentence to eliminate redundancy and improve clarity. The updated version combines the mention of CRISPR/Cas9 with its application in validating candidate genes, resulting in a more concise expression of the point.

Comments 18: References: Comprehensive and updated. Ensure the formatting is consistent (certain entries on Pages 13–17 are lacking periods or have inconsistent capitalization).

Response 18:

References have been checked and revised.

Comments 19: Language and Style: The language is scientific and fluent. Clarity may be achieved by breaking down occasional lengthy sentences (such as those on Pages 2, 5, and 11).

Response 19:

Thank you for your suggestions. I have revised the long sentences in the article.

Yours sincerely,

Dr. Weidong Deng and Xiaoming He

Yunnan Provincial Key Laboratory of Animal Nutrition and Feed, Faculty of Animal Science and Technology, Yunnan Agricultural University, Kunming 650201, China

College of Animal Husbandry and Veterinary Medicine, Yunnan Vocational College of Agriculture, Kun-ming 650201, China;

Tel.: +86-871-65220375

Email: dengwd@ynau.edu.cn(W.D.); xiaominghe@foxmail.com (X.H.)

Round 2

Reviewer 1 Report

Comments and Suggestions for Authors

I am not fully satisfied with the revision. I feel that the Authors have aimed for minimal changes and appear to be seeking ways to justify or explain away the issues rather than addressing them substantively. I feel that a more comprehensive approach to the revisions would better serve the work and address the concerns that were raised, especially in outlook part. 

I am still enthusiastic about the topic of this manuscript, as it addresses an important gap in the literature and has the potential to generate significant interest. Given the value of this research, I would like to request another round of revisions with my previous questions, as I believe that with more thorough development, this work could make a truly meaningful contribution to the field.

Author Response

Dear Reviewer,

We are grateful for the opportunity to revise our manuscript titled "Genomic and Molecular Mechanisms of Goat Environmental Adaptation" (Manuscript ID: biology-3652434). We appreciate the insightful comments and suggestions provided by the reviewers, which have significantly improved the quality of our work.

In response to the reviewers' feedback, we have made several key revisions to enhance the clarity, specificity, and impact of our manuscript:

Comments 1: The title is a bit overclaiming as the the content are limitedly discuss about the genome and epigenome studies in goat adaptation, except the ‘Overview of Goat Genomics Research’ part that is comprehensively discuss about the development of the goat genome. Thus, to make the title represents the whole article enough, I suggest to incorporate more discussion related to the topics.

Response 1:

Thanks the reviewer for the thoughtful comment regarding the scope of the title. We agree that while the article provides an in-depth overview of genomic mechanisms in goat environmental adaptation, the discussion on epigenomic aspects is comparatively brief. To ensure that the title accurately reflects the focus and content of the manuscript, we have revised it to: “Genomic and Molecular Mechanisms of Goat Environmental Adaptation”. This revised title better captures the balance of topics covered, avoids overstatement, and aligns with the article’s main themes. We sincerely appreciate the reviewer’s input, which helped us improve the clarity and consistency of the manuscript.

Comments 2: The number of reference is extensive, with the majority of them from the last decade publication.

Response 2:

We sincerely thank the reviewer for acknowledging the breadth and recency of the references cited. In this review, we made a conscious effort to include up-to-date studies from the past decade in order to accurately reflect the latest advances in goat genomics and environmental adaptation research.

Comments 3: In general, I can spot some inconsistency in italicizing the gene name, so please double check it. Additionally, some paragraph are lengthy, making it difficult to follow, so I suggest to break them down to some shorter ones.

Response 3:

Thanks to the reviewer for his suggestions, we will adjust and modify the italic parts of the whole text and split the longer paragraphs.

Comments 4: Simple summary: please pay attention for some terms that need to be italicized, such as the species scientific name and gene name.

Response 4:

Thank you for your attention to detail, so that we can make corrections.

Comments 5: The simple summary and abstract looks very similar. A simple summary aims to explain your work in a simple way for the public, but the abstract should provide a comprehensive review of the full article, including the important conclusions and not merely outlining the aspects explored in this work.

Response 5:

We thank the reviewer for pointing out the redundancy between the Simple Summary and the Abstract. We have thoroughly revised both sections to clarify their distinct purposes. The Simple Summary now presents a more concise and accessible overview of the study, intended for a general audience. In contrast, the Abstract has been rewritten to provide a comprehensive synthesis of the article, including major findings and conclusions, such as the identification of key genes (e.g., EPAS1, UCP1, HSP70), the integration of genomic and epigenetic mechanisms, and the implications for future precision breeding strategies. These changes help clearly differentiate the functions of both sections and enhance the clarity and impact of the manuscript.

Comments 6: On the last paragraph you mentioned about the integration of multi-approach genomics research to understand adaptation, however, very limited discussion about it is available in the following chapter.

Response 6:

We thank the reviewer for this suggestion. To this end, we will expand on the role of multi-omics integration in adaptive research strategies and deepen our understanding of complex adaptive traits in Section 5 of this paper. These additional content helps to clarify and strengthen the conclusions of the last paragraph while maintaining the overall structure and flow of the manuscript.

Comments 7: At the end of it, you mentioned about many unanswered scientific questions in this field, so I am expecting that these questions on the discussion part.

Response 7:

Thank you for this important suggestion from the reviewer. In the follow-up discussion, we summarized the relevant studies in recent years and answered the gene regulatory network of adaptive traits and the relationship between genomic and epigenetic regulation.

Comments 8: Subchapter 3.1.1. It is mainly discuss about adaptation mechanism in hot environment and limitedly discuss about the genetic make behind this mechanism. As the title of this article clearly emphasize the genetic/genomic aspect of environmental adaptation. I suggest authors to add more of this aspect to this subchapter.

Response 8:

We fully agree with the reviewer that the genetic basis of heat adaptation needs to be strengthened in Subchapter 3.1.1. Therefore, we added a concise paragraph highlighting key genes and genomic variants associated with thermotolerance in this section, such as HSP70 and ACTHR. These additions strengthen the genetic focus of this section while maintaining the overall flow of the Discussion, further leading to Section 3.3.

Comments 9: Subchapter 3.1.2. and 3.1.3. These parts are better than the previous one as we can see more genetics part in it. However, those parts that related to morphological/physiological adaptation mechanism should also be linked to the genetics background of the process.

Response 9:

We thank the reviewer for this constructive suggestion. In response, we have added brief statements in both Subchapter 3.1.2 (high-altitude adaptation) and 3.1.3 (cold adaptation) to explicitly link physiological traits—such as hemoglobin levels, alveolar changes, hair density, and metabolic rate—with genetic or transcriptomic evidence where available. These additions help integrate the morphological and physiological observations with underlying genetic mechanisms, strengthening the overall coherence of the discussion.

Comments 10: Table 1. It would be more informative if you can provide the year of the genome assembly versions created.

Response 10:

Thank you for your suggestion. We have added the genome assembly version time in Table 1.

Comments 11: Table 2. Among those methods, can you add 1-2 sentences prior to that table about which one is more commonly used or more informative?

Response 11:

Thank you for the helpful suggestion. We have added two sentences prior to Table 2 to briefly explain that population differentiation methods (e.g., Fst, XP-CLR) and haplotype-based methods (e.g., XP-EHH) are among the most widely used due to their sensitivity in detecting selection signatures under diverse evolutionary scenarios. This addition helps orient the reader and clarify the practical application of the listed methods.

Comments 12: You have mentioned about wool traits for adaptation in several part of the discussion, what about the coat/wool color role in the adaptation?

Response 12:

We appreciate the reviewer’s attention to this important yet under-discussed aspect. We have added a brief paragraph to Section 3.1.1 to discuss the role of coat color in adaptation, particularly in relation to solar radiation and thermoregulation. We also mention the involvement of pigmentation-related genes such as KITLG and ASIP, which have shown signs of selection in desert-adapted goat populations. This addition complements the discussion on wool structure and enhances our coverage of integumentary traits.

Comments 13: Figures: improve the quality, as they are a bit blurry, especially Figure 1.

Response 13:

Thank you very much for your suggestion. We will insert the image into a vector image to improve the image clarity.

Comments 14: Subchapter 3.3. The suggestion about further integration of genomics analysis and so on are too general. I would expect you to mention one of two specific ideas for this future direction of the study.

Response 14:

We appreciate the reviewer’s comment. To address this, we have revised Subchapter 3.3 to include specific proposals for future research directions. These include conducting environmental GWAS in goat populations from contrasting habitats (e.g., high-altitude vs. tropical lowlands) and applying integrative network analysis to link selected loci with phenotypic and environmental data. These examples provide actionable frameworks for advancing the study of environmental adaptation.

Comments 15: If it is possible, I would also suggest to add one more subchapter regarding the gene editing research on goats environmental adaptation, if related researches are available.

Response 15:

We sincerely thank the reviewer for the thoughtful suggestion regarding gene editing in goats. While we fully agree that CRISPR/Cas9 and related technologies offer promising avenues for future research, current studies specifically linking gene editing to environmental adaptation in goats remain limited. We added two examples of goat gene editing at the end of 3.3 to help us better introduce subsequent suggestions for goat breeding and farming.

Comments 16: For the outlook part, it sounds very general, again, I expect you to mention one or two specific ideas on how can better comprehend the understanding of the environmental adaptation in goats, e.g., conducting a GWAS for animal in two extreme living condition or incorporating the wild relatives in the study to compare its adaptation capability, or anything else.

Response 16:

Thank you for the reviewer's valuable suggestions. Therefore, we added a section before the conclusion and outlook on specific ideas for understanding the environmental adaptability of goats, including environmental gradient-phenotype association studies and multi-omics dynamic monitoring and functional verification. In addition, we have made more specific revisions to the outlook, and we welcome your continued suggestions.

Yours sincerely,

Dr. Weidong Deng and Xiaoming He

Yunnan Provincial Key Laboratory of Animal Nutrition and Feed, Faculty of Animal Science and Technology, Yunnan Agricultural University, Kunming 650201, China

College of Animal Husbandry and Veterinary Medicine, Yunnan Vocational College of Agriculture, Kun-ming 650201, China;

Tel.: +86-871-65220375

Email: dengwd@ynau.edu.cn(W.D.); xiaominghe@foxmail.com (X.H.)